# Pertuzumab and Trastuzumab Combination with Concomitant Locoregional Radiotherapy for the Treatment of Breast Cancers with HER2 Receptor Overexpression

**DOI:** 10.3390/cancers13194790

**Published:** 2021-09-24

**Authors:** Amélie Aboudaram, Pierre Loap, Delphine Loirat, Syrine Ben Dhia, Kim Cao, Alain Fourquet, Youlia Kirova

**Affiliations:** 1Department of Radiation Oncology, Institut Curie, 75005 Paris, France; amelie.aboudaram@curie.fr (A.A.); pierre.loap@curie.fr (P.L.); syrine.bendhia@curie.fr (S.B.D.); kim.cao@curie.fr (K.C.); alain.fourquet@curie.fr (A.F.); 2Department of Medical Oncology, Institut Curie, 75005 Paris, France; delphine.loirat@curie.fr

**Keywords:** breast cancer, concurrent pertuzumab-trastuzumab-radiotherapy

## Abstract

**Simple Summary:**

This retrospective study demonstrates that the combination of locoregional breast RT with dual HER2 blockade by pertuzumab/trastuzumab was very well tolerated, suggesting that RT can be safely administered to patients with HER2-positive breast cancer during dual HER2 blockade. Excellent locoregional control was achieved in irradiated patients.

**Abstract:**

Background: The combination of pertuzumab and trastuzumab dual HER2 blockade with concomitant curative dose locoregional breast radiotherapy in patients with metastatic breast cancer is an important part of treatment strategy. Methods: This was a retrospective study conducted at the Institut Curie on all patients treated concomitantly with pertuzumab/trastuzumab and locoregional breast radiotherapy. Toxicity was evaluated according to the NCICTCAEv4.0. Overall survival, progression-free survival and locoregional recurrence-free survival were evaluated in metastatic patients who were initially well controlled by chemotherapy, for whom local treatment was decided by the multidisciplinary team. Results: Fifty-five patients treated between October 2013 and December 2019 were included, with a median follow-up of 4.1 years. The median age was 53 years (range: 28–81). All patients received curative dose radiotherapy (RT) concomitantly with pertuzumab and trastuzumab (Pertu/Trastu). The median radiation dose was 50 Gy. Safety evaluation did not reveal any significant adverse effects, with 3 cases of grade 3 radiodermatitis (5.4%), but no significant gastrointestinal or cardiac toxicity. The mean difference in LVEF before any chemotherapy and after radiotherapy was −2.43% (*p* < 0.01). Conclusions: This study demonstrates that the combination of locoregional breast RT with dual HER2 blockade by Pertu/Trastu was very well tolerated, suggesting that RT can be safely administered to patients with HER2-positive breast cancer.

## 1. Introduction

HER2 receptor (Human Epidermal growth factor Receptor 2) overexpression is present in 15 to 30% of all breast cancers [1]. This receptor is located on chromosome 17q21 and is activated by dimerization or cleavage of its extracellular domain, which triggers a signalling cascade in the MAP kinase pathway, or the PI3K/Akt/MTOR system, generating cell proliferation and promoting division and growth of cancer cells [2].

Trastuzumab is the first humanized monoclonal antibody targeting the HER2 receptor [3], directed against the extracellular domain of the receptor, thereby blocking the MAP kinase signalling pathways and the PI3K-Akt pathway, which slows the cell cycle and decreases tumour cell proliferation [3]. Trastuzumab also allows recruitment of innate immune cells, macrophages and NK (natural killer) cells [3,4].

The efficacy of trastuzumab in HER2-positive breast cancer has been demonstrated at all stages of disease [5,6].

Pertuzumab is also a humanized anti-HER2 monoclonal antibody [7]. Pertuzumab specifically binds to the extracellular dimerization domain of the HER2 protein and blocks ligand-dependent heterodimerization of HER2 and other HER family receptors, including EGFR (Epidermal Growth Factor Receptor), HER3, and HER4, thereby inhibiting ligand-dependent intracellular signalling pathways, leading to arrest of cell proliferation and apoptosis [1].

The combined action of these two antibodies reinforces blockade of the transmission of cancer cell proliferation and survival signals [8]. Up until now, this treatment has been reimbursed by French national health insurance only for first-line treatment of metastatic disease in patients not previously treated with trastuzumab.

Various combinations of the two anti-HER2 antibodies with concomitant radiotherapy can be proposed: for analgesic or decompressive irradiation in advanced metastatic disease, for local chest wall or breast irradiation, or locoregional irradiation of lymph nodes in metastatic patients in complete response, in whom consolidation radiotherapy is envisaged.

Although the efficacy of these two treatments has been clearly demonstrated, only limited data are available concerning the safety and toxicity of this combination in conjunction with radiotherapy.

A recently published study demonstrated the safety of the combination of dual HER2 blockade and radiotherapy in various palliative or consolidation radiotherapy indications [9]. We decided to review this patient cohort, the longest cohort currently available to our knowledge, in order to complete these data and to validate the safety of dual HER2 blockade in combination with curative dose radiotherapy.

## 2. Material and Methods

### 2.1. Study Population

The patient cohort was derived from the single-centre retrospective study by Ben Dhia et al. [9] on all patients treated at Institut Curie treated by the combination of pertuzumab–trastuzumab dual HER2 blockade and a concomitant curative dose of locoregional radiotherapy to the breast or chest wall and/or adjacent lymph nodes.

For all patients, to identify HER2, amplification was performed immunohistochemistry (IHC) and in case of a HER2++ result, HER2 amplification status was determined by commercially available FISH.

Patients of this cohort with breast cancer and HER2 receptor overexpression were treated with dual HER2 blockade combining trastuzumab at a loading dose of 8 mg/kg, followed by a monthly maintenance dose of 6 mg/kg, and pertuzumab at a loading dose of 840 mg followed by a maintenance dose of 420 mg. Concomitant chemotherapy was also administered in certain cases. All patients underwent a pretreatment cardiac assessment including determination of left ventricular ejection fraction (LVEF), followed by LVEF monitoring every 12 to 16 weeks.

The present analysis was based on patients receiving local radiotherapy (breast or chest wall and/or to supradiaphragmatic lymph nodes ipsilateral to the primary tumour), at a minimum biological equivalent dose (EQD2) of 40 Gy. The study flowchart is shown in Figure 1.

The choice of irradiation technique (3D conformal radiotherapy [10], intensity-modulated conformal radiotherapy [11], arc therapy), the treatment device and the type of radiation used (photons or electrons [12]) were adapted to each patient’s anatomy and complied with the Institut Curie Department of Radiation Oncology technical guidelines. The boost dose to the tumour bed, when administered, used photons or electrons depending on the tumour site and the technique used and the volume was defined according to the Institut Curie protocol [13,14].

We defined acute toxicity as any adverse event occurring from the beginning to 3 months after completion of radiotherapy, and late toxicity as any adverse event occurring more than 3 months after radiotherapy. Toxicity was evaluated and graded according to the National Cancer Institute Common Terminology Criteria for Adverse Events version 4.0 (NCI-CTCAE).

This retrospective study was accepted by the Breast Cancer Research and Treatment Group.

### 2.2. Statistical Analysis

Qualitative data were expressed as number and percentage, quantitative data were expressed as the median and range. Pretreatment and post-radiotherapy LVEF values were compared by a paired Student’s test; the normal distribution of variables was verified by a Jarque–Bera test. The limit of statistical significance was set as *p* < 0.05.

For the subgroup of patients with metastatic disease at the time of diagnosis, overall survival, progression-free survival and locoregional recurrence-free survival were defined from the date of completion of radiotherapy; data were censored in April 2021. Survival data were analysed using the Kaplan–Meier method; hazard ratios were calculated using the Cox proportional hazards model. All statistical analyses were carried out with R software version 3.6.1 (https://www.r-project.org/, accessed on 23 September 2021).

## 3. Results

### 3.1. Study Population

Of the 77 patients initially irradiated concomitantly with dual HER2 blockade, we excluded 22 patients who received palliative doses of irradiation, all sites combined, i.e., doses less than 40 Gy EQD2. Of the remaining patients, 37 patients had metastatic disease at the time of diagnosis, 8 patients subsequently developed metastases after initial treatment, and 10 patients had only locally or locoregionally advanced disease (Figure 1).

Patient characteristics are presented in Table 1. The median age of the patients was 53 years (range: 28 to 81 years). For the vast majority of patients (94.5%), tumour histology consisted of infiltrating ductal carcinoma. Patients mostly presented advanced disease with 58.2% stage T3-4, 58.2% stage N+, and high-grade tumours, with 58.2% grade 3. Four patients (7.3%) had a history of cardiovascular disease prior to radiotherapy.

### 3.2. Treatment

#### 3.2.1. Systemic Therapy

At the time of initiation of pertuzumab/trastuzumab dual HER2 blockade, all patients had previously received chemotherapy based on docetaxel for 69.1% of patients, paclitaxel for 9.1% of patients and epirubicin for 14.5% of patients. One patient had received both epirubicin and docetaxel (the chemotherapy regimen was not specified for 5 patients).

The median dose received was 14,787 mg (4400–37,340) for trastuzumab and 13,860 mg (4200–42,508) for pertuzumab.

#### 3.2.2. Surgery

A total of 10 patients (18.2%) were not operated on, 23 patients (41.8%) underwent lumpectomy and 19 patients (34.5%) underwent mastectomy. A chest wall metastasis was resected in 3 patients (5.5%) in the context of distant recurrence after upfront mastectomy.

#### 3.2.3. Concomitant Radiotherapy

The median dose received was 50 Gy EQD2 (range: 40 Gy–74 Gy). Irradiated volumes corresponded to the chest wall for 23 patients (after mastectomy or chest wall metastasectomy) or the breast for 30 patients and 15 patients received a boost to the tumour bed. Two patients with ipsilateral lymph node metastatic recurrence only received regional irradiation. A total of 42 patients (76.4%) received regional lymph node radiotherapy. Radiotherapy was delivered concomitantly with pertuzumab/trastuzumab dual HER2 blockade.

### 3.3. Toxicity and Safety

#### 3.3.1. Skin Toxicity

Early toxicity consisted of 3 cases (5.4%) of grade 3 and 14 cases (25.5%) of grade 2 spontaneously resolving radiation dermatitis. The remaining patients experienced grade 0 and grade 1 reactions (69.1%). One case (1.8%) of grade 2 telangiectasia was observed, corresponding to late toxicity.

#### 3.3.2. Gastrointestinal Toxicity

One case (1.8%) of grade 1 early oesophagitis was observed during radiotherapy.

#### 3.3.3. Cardiac Toxicity

No cardiac events were observed during concomitant therapy. An episode of myocardial ischaemia with normal coronary angiography was observed 9 months after treatment in a patient receiving anthracyclines at the time of the event. The impact on LVEF was reversible when chemotherapy was discontinued. The causal role of radiotherapy therefore appears to be unlikely.

LVEF before initiation of chemotherapy and LVEF after radiotherapy were available for all patients of the cohort. The observed LVEF differences therefore occurred after all treatment modalities, particularly anthracycline-based chemotherapy and HER2 blockade.

A statistically significant 2.43% (−17–+14%) decrease in LVEF was observed between the start and end of treatments. No patient simultaneously experienced a greater than 10% decrease in LVEF associated with LVEF <55%. No significant difference in the decrease of LVEF was observed according to the side treated: post-left RT LVEF = 65% (55–73%) vs. post-right RT LVEF = 65% (48–77%) (*p* = 0.39).

#### 3.3.4. Other Toxicities

No lung or thyroid toxicity was reported.

### 3.4. Survival Data

Survival data were calculated in the 37 patients with metastatic disease at the time of diagnosis, corresponding to the largest subgroup. This approach was adopted in order to ensure a more homogeneous cohort, as patients with non-metastatic breast cancer have long survival and the follow-up of our cohort was not sufficient to assess this variable. Furthermore, the pertuzumab/trastuzumab combination is not part of the standard treatment of non-metastatic patients and is not reimbursed by French national health insurance.

The characteristics of the population with metastatic disease at the time of diagnosis are presented in Table 2. The median age of the patients at the time of diagnosis was 51 years (28–68). The vast majority of patients (94.6%) had no history of cardiovascular disease. The predominant histology was ductal carcinoma in situ (97.3%) with predominantly high histological grade (59.5% of grade 3). Various metastatic sites were observed, i.e., bone (24.3%), liver (18.9%), lung (16.2%); 35.1% of patients had multiple metastases.

Median follow-up after radiotherapy was 4.1 years. Four-year overall survival (OS) was 81.7% (69.5%−96.2%) (Figure 2). Progression-free survival (PFS) was 65.7% (51.7%−83.6%) (Figure 3). Locoregional recurrence-free survival (LRRFS) was 97.1% (91.5%−100%) (Figure 4). Median OS, PFS and LRRFS were not reached.

## 4. Discussion

Data from our cohort demonstrate the excellent safety of concomitant locoregional RT and dual HER2 blockade in terms of skin, gastrointestinal and general toxicity. No significant cardiac toxicity was observed, apart from a slight decrease in LVEF, which is expected during HER2 blockade [15].

This is the first cohort to assess overall survival and 4-year progression-free survival associated with the combination of RT and pertuzumab/trastuzumab dual HER2 blockade. This study demonstrates an excellent locoregional control rate, and an overall survival that seems to be consistent with studies evaluating the survival of patients with irradiated metastatic HER2-positive breast cancer [8].

The efficacy of locoregional treatment for patients with metastatic disease was demonstrated in the ESME cohort [16].This series included more than 4500 patients with metastatic breast cancer at the time of diagnosis. A subgroup of 1965 patients with no disease progression after one year of systemic therapy was identified. Within this subgroup, 45% of patients received locoregional treatment consisting of exclusive surgery (13.7%), exclusive external beam radiotherapy (41.1%) or a combination of surgery and radiotherapy (45.2%), and 55% did not receive any local treatment. Overall survival after a median follow-up of 45.6 months was 65 months. Multivariate analysis demonstrated a reduction of mortality of 37% (HR [hazard ratio]: 0.63) and 39% (HR: 0.61) in the two local treatment subgroups receiving exclusive radiotherapy or a combination of surgery and radiotherapy, respectively, compared to patients without local treatment, but no reduction of mortality was observed in patients treated by surgery alone, confirming the major role of locoregional treatment for metastatic disease in selected patient groups [16,17].

The pertuzumab and trastuzumab combination enhances antitumour activity and has been clearly shown to be effective in the management of HER2-positive breast cancer at different stages of the disease [18].

Dual HER2 blockade can be combined with radiotherapy, even at curative doses higher than 40 Gy EQD2. Personalized locoregional radiotherapy using innovative irradiation techniques ensures good safety of treatment [10,11,12,13].

On the basis of this study, which, to the best of our knowledge, is the largest series to evaluate the safety of this combination, we can conclude the absence of any added toxicities with a good safety profile, especially in terms of cardiac toxicity and a positive benefit/risk balance.

## 5. Conclusions

This study demonstrates that the combination of locoregional breast RT with dual HER2 blockade by pertuzumab/trastuzumab was very well tolerated, suggesting that RT can be safely administered to patients with HER2-positive breast cancer during dual HER2 blockade. Excellent locoregional control was achieved in irradiated patients. Survival data appear to be comparable those reported in the literature for HER2-positive metastatic breast cancer patients on dual HER2 blockade.

## Figures and Tables

**Figure 1 cancers-13-04790-f001:**
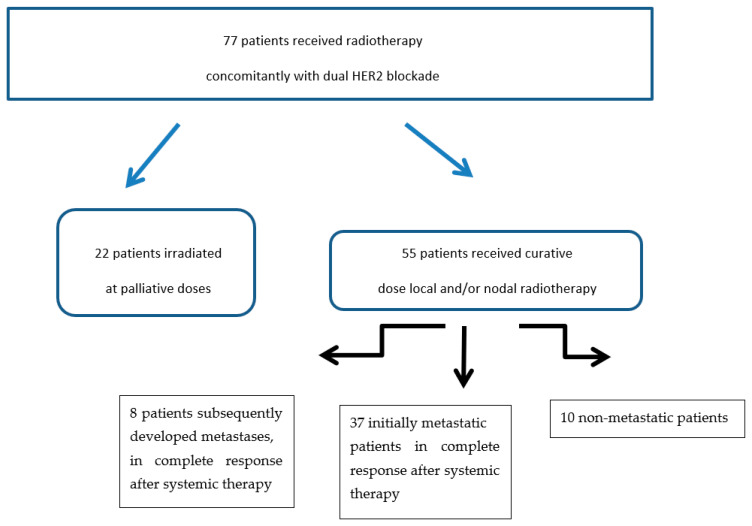
Flowchart.

**Figure 2 cancers-13-04790-f002:**
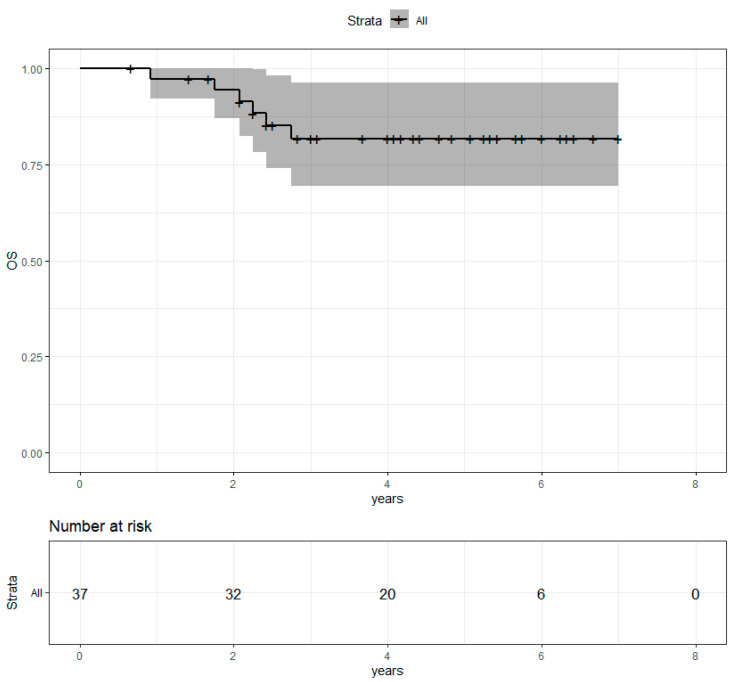
Overall survival of the population with metastatic disease at the time of diagnosis after concomitant RT and Pertu/Trastu dual HER2 blockade. Confidence intervals are provided in grey.

**Figure 3 cancers-13-04790-f003:**
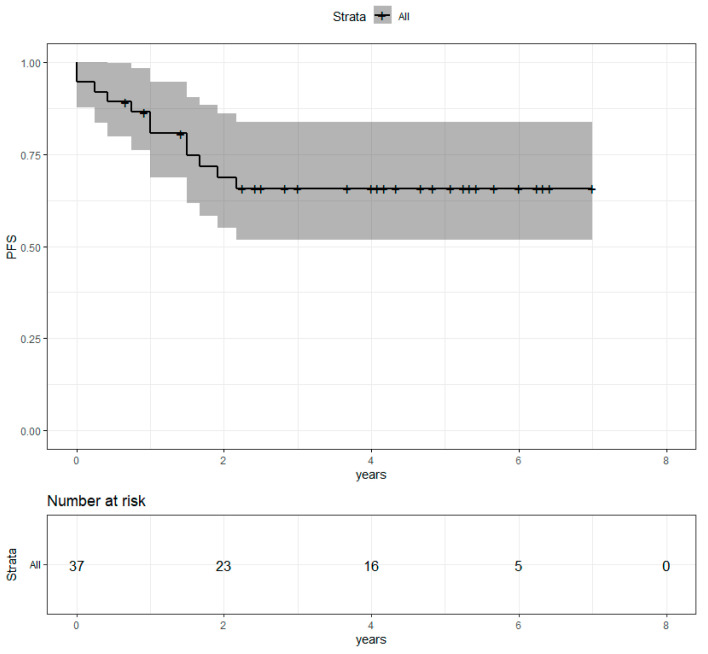
Progression-free survival of the population with metastatic disease at the time of diagnosis after concomitant RT and Pertu/Trastu dual HER2 blockade. Confidence intervals are provided in grey.

**Figure 4 cancers-13-04790-f004:**
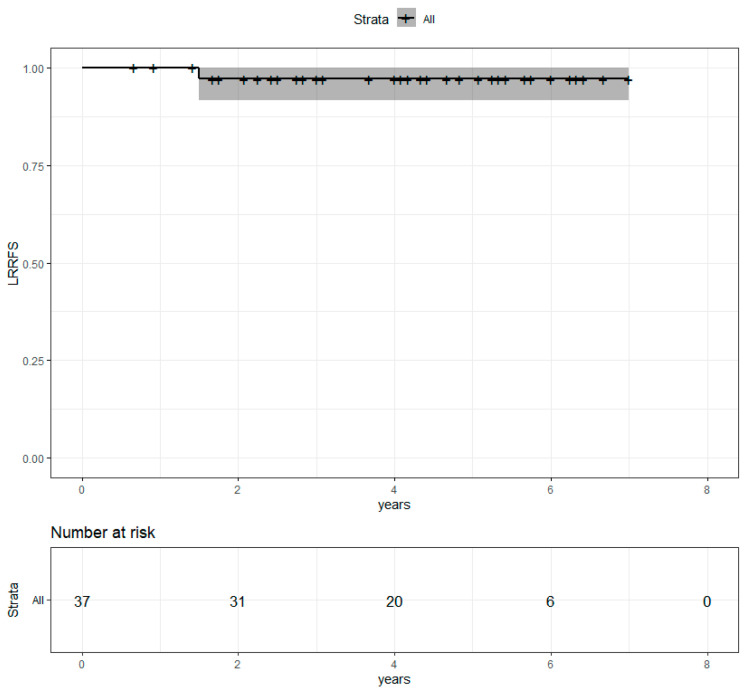
Locoregional recurrence-free survival in the population with metastatic disease at the time of diagnosis after concomitant RT and Pertu/Trastu dual HER2 blockade. Confidence intervals are provided in grey.

**Table 1 cancers-13-04790-t001:** Study population (patients received curative dose local and/or lymph nodes radiotherapy.

Characteristics		N (55)	%
**Age (years)**	53 (28–81)		
**Cardiovascular history**			
Yes		4	7.3
No		51	92.7
**BMI**	23.1 (18.3−35.6)		
**Primary tumor laterality**			
Left		21	38.2
Right		31	56.4
Bilateral		3	5.5
**Histological type**			
Ductal		52	94.5
Lobular		2	3.6
Other		1	1.8
**Histological grade**			
2		20	36.4
3		32	58.2
NA		3	5.5
**Stage (at baseline)**			
T3-T4		32	58.2
N+		32	58.2
M+		35	63.6
**Context of locoregional irradiation with Trastuzumab + Pertuzumab**			
Metastatic (at diagnosis)		37	67.3
Metastatic (metachrone)		8	14.5
Non metastatic setting		10	18.2

**Table 2 cancers-13-04790-t002:** Characteristics of the population with metastatic disease at the time of diagnosis.

Characteristics		N (37)	%
**Age (years)**	51 (28–68)		
**Cardiovascular history**			
Yes		2	5.4
No		35	94.6
**BMI**	23.4 (18.3–34.0)		
**Primary tumor laterality**			
Left		11	29.7
Right		24	64.9
Bilateral		2	5.4
**Histological type**			
Ductal		36	97.3
Lobular		1	2.7
**Histological grade**			
2		13	35.1
3		22	59.5
NA		2	5.4
**Metastatic site**		37	100
Bone		9	24.3
Hepatic		7	18.9
Lung		6	16.2
Other organ		2	5.4
Multiple organs		13	35.1
**Locoregional extension**			
T3-T4		23	62.2
N+		22	59.5

## Data Availability

In case of additional questions, the data are available.

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
