# Peer review of "Pertuzumab and Trastuzumab Combination with Concomitant Locoregional Radiotherapy for the Treatment of Breast Cancers with HER2 Receptor Overexpression"

_cancers, 2021, doi:10.3390/cancers13194790_

Round 1
Reviewer 1 Report
In a present retrospective study, the authors examine 55 patients treated with RT and Her2 blockade by Pertu/Trastu and infer that it is well tolerated with RT. I have several reservations, my comments are appended as below:
- In the first place, authors should explain what ‘locoregional’ means for a general audience.
- The study includes fewer patients. The authors should try to look for the datasets available with similar characteristics and try to validate the observations.
- Study population: please explain first how the diagnosis was carried out.
- Do authors study the pre and postmenopausal women and responsiveness to therapy?
- Authors should include the representative HE images.
- Survival curves: please annotate with statistical inference.
- They should compare the survival between two groups, otherwise, it’s hard to judge the merit of present therapy.
Author Response
- In the first place, authors should explain what ‘locoregional’ means for a general audience.
- The authors are thankful for this remark, please find in the text:
The present analysis was based on patients receiving local radiotherapy (breast or chest wall) and/or to supradiaphragmatic lymph nodes ipsilateral to the primary tumour, at a minimum biological equivalent dose (EQD2) of 40 Gy.
The choice of irradiation technique (3D conformal radiotherapy10, intensity-modulated conformal radiotherapy11, arc therapy), the treatment device and the type of radiation used (photons or electrons12) were adapted to each patient's anatomy and complied with the Institut Curie Department of Radiation Oncology technical guidelines. The boost dose to the tumour bed, when administered, used photons or electrons depending on the tumour site and the technique used and the volume was defined according to the Institut Curie protocol13,14.
- The study includes fewer patients. The authors should try to look for the datasets available with similar characteristics and try to validate the observations.
- This is very pertinent question but unfortunately there is no other data asking the same question. Therefore these series are uniques. Please find in the text:
This is the first cohort to assess overall survival and 4-year progression-free survival associated with the combination of RT and pertuzumab/trastuzumab dual HER2 blockade. This study demonstrates an excellent locoregional control rate, and an overall survival that seems to be consistent with studies evaluating the survival of patients with irradiated metastatic HER2-positive breast cancer8.
- Study population: please explain first how the diagnosis was carried out.
- Done, Please find in the text:
For all patients to identify HER2 amplification was performed immunohistochemistry (IHC) and in case of HER2++ result, HER2 amplification status was determined by commercially available FISH.
- Do authors study the pre and postmenopausal women and responsiveness to therapy?
- Unfortunately this question was not asked because the small number of premenopausal patients, some perimenopausal
- Authors should include the representative HE images.
- Thank you for this suggestion but this is not the subject of the study
- Survival curves: please annotate with statistical inference.
- Done
- and for information: median OS, PFS and LRRFS were not reached (addede in the text)
- They should compare the survival between two groups, otherwise, it’s hard to judge the merit of present therapy.
- Unfortunately this is not randomized and there is not enough statistical power to do this kind of comparison
Reviewer 2 Report
After carefully read the manuscript, I do not have concerns. The article is written in a readily accessible manner and it is suitable for this journal.
Author Response
The authors are very thankful to reviewer for this excellent understanding of our work.
Reviewer 3 Report
Overall, the manuscript was well written and described the relevant background that was informative for the cohort that was evaluated in the retrospective analysis. The authors sufficiently described the cohort in Table 1. The rationale for evaluating 37 metastatic patients for the survival analysis was clearly described. In the discussion, the authors provided context for the rationale and value of evaluating the dual HER2 blockade metastatic breast cancer patients. Overall, I consider this manuscript to be appropriate for publication.
There are minor edits listed below.
- Page 4, Table 1- There are some words in French.
- Under histological type, change "autre" to "other"
- change "Stade" to "Stage"
Author Response
Overall, the manuscript was well written and described the relevant background that was informative for the cohort that was evaluated in the retrospective analysis. The authors sufficiently described the cohort in Table 1. The rationale for evaluating 37 metastatic patients for the survival analysis was clearly described. In the discussion, the authors provided context for the rationale and value of evaluating the dual HER2 blockade metastatic breast cancer patients. Overall, I consider this manuscript to be appropriate for publication.
The authors are very thankful to reviewer for this professional understanding of our work
There are minor edits listed below.
- Page 4, Table 1- There are some words in French.
- Under histological type, change "autre" to "other"
- change "Stade" to "Stage"
Done
Round 2
Reviewer 1 Report
I congratulate the authors for the modifications. I suggest accepting now.